biomechanics/plant science

bees, biomechanics, buzz pollination, heteranthery, pollination, *Solanum*

**Author for correspondence:**
Vinicius Lourenço Garcia Brito
e-mail: viniciusduartina@gmail.com

# Biomechanical properties of a buzz-pollinated flower

Vinicius Lourenço Garcia Brito[1],

Carlos Eduardo Pereira Nunes[2], Caique Rocha Resende[1],

Fernando Montealegre-Zapata[3] and

Mario Vallejo-Marín[2]

[1]Instituto de Biologia, Universidade Federal de Uberlândia, Uberlândia, MG 38405-315 Brazil
[2]Department of Biological and Environmental Sciences, University of Stirling, Stirling FK9 4LA, UK
[3]School of Life Sciences, University of Lincoln, Lincoln LN67DL, UK

VLGB, 0000-0002-2727-3063; CEPN, 0000-0002-7534-0697; FM-Z, 0000-0001-5186-2186; MV-M, 0000-0002-5663-8025

Approximately half of all bee species use vibrations to remove pollen from plants with diverse floral morphologies. In many buzz-pollinated flowers, these mechanical vibrations generated by bees are transmitted through floral tissues, principally pollen-containing anthers, causing pollen to be ejected from small openings (pores or slits) at the tip of the stamen. Despite the importance of substrate-borne vibrations for both bees and plants, few studies to date have characterized the transmission properties of floral vibrations. In this study, we use contactless laser vibrometry to evaluate the transmission of vibrations in the corolla and anthers of buzz-pollinated flowers of *Solanum rostratum*, and measure vibrations in three spatial axes. We found that floral vibrations conserve their dominant frequency (300 Hz) as they are transmitted throughout the flower. We also found that vibration amplitude at anthers and petals can be up to greater than 400% higher than input amplitude applied at the receptacle at the base of the flower, and that anthers vibrate with a higher amplitude velocity than petals. Together, these results suggest that vibrations travel differently through floral structures and across different spatial axes. As pollen release is a function of vibration amplitude, we conjecture that bees might benefit from applying vibrations in the axes associated with higher vibration amplification.

## 1. Introduction

Vibrations play an important role in diverse biological interactions involving plants and animals [1–3]. Communication in invertebrates often occurs through vibrations that are transmitted through the substrate, particularly plant structures [4,5]. For example, male

and female wandering spiders use plant leaves to detect each other during pre-copulation and some hemipteran predators can detect vibrations produced by leaf-feeding caterpillars during prey search [6,7]. Leaf-cutting ants use stridulation to recruit nest-mates, and dry wood termites use resonant frequencies to assess the size of a given wood block [8,9]. In these cases, substrate properties can affect the vibrations and mediate information transmitted from sender to receiver. To date, our understanding of how plant structures modify the transmission of vibrations is relatively limited [3,5].

Beyond communication, vibrations are also involved in another extraordinary interaction between invertebrates and plants. Some insects, specifically bees, use vibrations to extract pollen grains from certain types of flowers, in a phenomenon called floral buzzing or sonication, which gives rise to the buzz pollination syndrome [10–13]. Buzz pollination has evolved independently multiple times across flowering plants, being found in more than 65 plant families [10,12]. Although the morphology of buzz-pollinated flowers ranges widely [12,14], many buzz-pollinated flowers have repeatedly converged to similar morphologies (e.g. the *Solanum*-like flower type), with anthers that dehisce through small apical pores (poricidal anthers) arranged in a cone-like central structure, as exemplified by some species in genus *Dodecatheon* (=*Primula*, Primulaceae), *Miconia* (Melastomataceae), *Solanum* (Solanaceae) and many others [15–17]. In buzz-pollinated flowers, mechanical vibrations produced by the thoracic muscles of visiting bees are transmitted to floral tissues resulting in pollen ejection via the apical pores [10,12]. Understanding how bee vibrations are transmitted through different floral structures has the potential to illuminate the mechanistic function of buzz-pollinated flowers while providing the background for further ecological and evolutionary studies of buzz pollination from both bee and flower perspectives.

The behaviour of bees while producing floral vibrations is relatively stereotypical [12,18]. Usually, bees land on the flower and use their mandibles and legs to grasp one or multiple stamens, after which they begin to rapidly contract their thoracic flight muscles transmitting vibrations to the whole flower [2,19]. Although bees may apply vibrations only to one stamen, pollen can be released from all stamens in the flower simultaneously. This effect is perhaps more dramatically demonstrated in flowers that possess two or more sets of morphologically distinct types of stamens, i.e. they are heterantherous [20–22]. In these heterantherous flowers, bees usually focus on their vibration efforts into only one set of stamens, the feeding anthers, while the second set of stamens, the pollinating anthers, deposits pollen on a different part of the bee's body disproportionally contributing to plant fertilization [23–25]. Therefore, when bees collect pollen directly from centrally located feeding anthers, the vibrations applied there need to be simultaneously transmitted to the pollinating anthers to enable pollen release. In heterantherous species, the transmission of vibrations from one part of the flower to another has immediate and important consequences for their reproductive success.

In buzz-pollinated flowers of the *Solanum*-type, which include some heterantherous species such as the one studied here (*Solanum rostratum*), individual or fused petals form sheet-like, flattened and elastic structures that are projected perpendicular or reflexed from the central axis of the flower (figure 1). By contrast, the stamens have a cylindrical or conical shape and are made of relatively rigid tissue, projected forward in parallel to the central axis of the flower (figure 1). The different configuration of petals and anthers raises the hypothesis that petals and anthers might differ in the magnitude of vibrations experienced across different axes of the flower. In a qualitative sense, petals and stamens can be roughly thought of as bending cantilever structures [3,26] that have a fixed end attached to the base of the flower and are subject to possible transversal and longitudinal bending vibrations applied by bees. In cantilever bending, the displacement of the structure's free end is directly proportional to the cube of its length, and inversely proportional to the material stiffness and its second moment of area [27]. Feeding and pollinating stamens have different sizes and shapes, suggesting that they might also have different vibrational properties [22]. Therefore, we predict that in heterantherous flowers such as the one studied here, longer (pollinating) stamens should show higher vibration amplitudes at their tip during buzz pollination compared to shorter (feeding) stamens.

In this study, we aim to characterize the transmission of vibrations in different floral organs in buzz-pollinated plants. Specifically, we address two questions: (i) Do the differences in size and morphology of feeding and pollinating anthers translate into different capacities for transmitting vibrations? (ii) Do petals and anthers differ in their transmission of vibrations across different axes of vibration (x, y and z in figure 1*b*)? We address these questions by applying artificial vibrations to the base of the flower (receptacle) and recording vibrations in petals or anther tips. Addressing these questions will help us understand how floral structures are more or less prone to vibrate during buzz pollination, and how it could ultimately affect pollen release by anthers. Our study is the first to investigate vibration transmission across multiple flower organs and multiple axes of vibration.

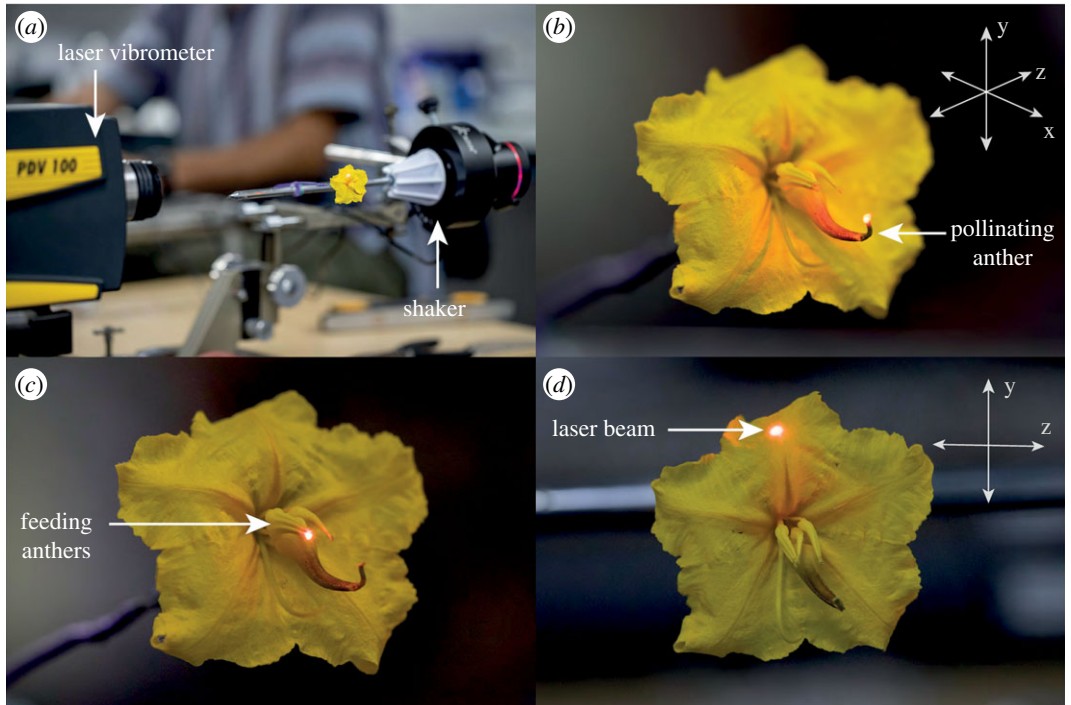

**Figure 1.** Experimental set-up for contactless measurement of the transmission of vibrations through the flowers of a buzz-pollinated plant, *S. rostratum* (Solanaceae). (*a*) Experimental set-up. The first laser vibrometer (PDV 100) is shown in the foreground, with the laser beam hitting the upper petal of the flower along the z-axis. The vibration transducer system is parallel to the laser beam and is shown in the background. The second laser is aimed at the flower receptacle behind the petals also along the z-axis; the second laser vibrometer is outside of the frame. The three measured axes (x, y and z) are shown as arrows at the top-right of (*b*) and (*d*). The perspective of the x and z-axes in (*b*) is exaggerated for illustration purposed. (*b*), (*c*) and (*d*) show the laser beam hitting the pore of the pollinating anther (*b*), the pore of the feeding anther (*c*) or the upper petal (*d*) with the laser beam parallel to the z-axis.

# 2. Material and methods

## 2.1. Study system

We used *Solanum rostratum* Dunal (Solanaceae), a buzz-pollinated annual herb with pollen-flowers, i.e. flowers that offer only pollen as resource for pollinators (figure 1) [20,28]. The genus *Solanum* includes more than 1300 species of buzz-pollinated plants and has a long tradition in the study of buzz pollination [29,30]. The pollen grains are concealed in anthers that dehisce only through a small pore in their apex (poricidal anthers). The flowers of *S. rostratum* are heterantherous, with two morphologically and functionally distinct sets of stamens in the same flower [25]. One set consists of four anthers (feeding anthers) that are small, bright yellow and centrally located within the flower. The second set consists of a single anther (pollinating anther) that is large, yellow to brown and positioned opposite to the deflected style [31]. Some bees visiting *S. rostratum* use vibrations to extract and collect pollen grains (floral vibrations; [28,32]). During floral visitation, bees curve their body around the anthers and vibrate their thoracic muscles [13]. In general, pollen release from flowers with poricidal anthers is related to the vibrational properties of floral vibrations [15,33,34].

## 2.2. Plant growth and flower material

Plants of *S. rostratum* were grown from seeds collected in 2010 from a population near San Miguel De Allende, Queretaro, Mexico (20.902° N, 100.706° W, 2033 m.a.s.l.; accessions 10s77, 10s79, 10s81 and 10s86). Seeds were germinated and plants grown as described by Vallejo-Marín *et al.* [17]. Plants (1–3 per accession) were transplanted either to 1.5 l pots or onto flowerbeds (1 × 5 m, with plants spaced about 50 cm from each other) in the research glasshouses at the University of Stirling. Plants were grown with 16 h of supplemental lighting a day and supplemental heating (25°/16°C, day/night).

Experimental flowers (unvisited by pollinators) were collected early in the morning (2–3 h after flower opening) by cutting an entire inflorescence and placing it in flower foam (Oasis Floral Products, Washington, UK) and transported to the laboratory in a plastic container. In the laboratory, the plastic container was kept open in a room with controlled temperature (21°C) and humidity (50% RH). For each trial, we used a single flower with the pedicel cut immediately underneath the receptacle.

## 2.3. Generation and playback of vibration signals

To study the biomechanical transmission of vibrations through flower structures, we synthesized vibration signals using Audacity (www.audacityteam.org). We used sine waves with 300 Hz frequency to capture the fundamental frequency of vibrations that bumblebees (*Bombus terrestris*) produce on flowers of *S. rostratum* based on previous studies with the same study system [33,35,36]. The synthesized vibrations had a maximum relative amplitude of one, the sampling rate of 44.1 kHz and phase $\varphi = 0°$. Using a computer, the vibrational signals produced were played in a custom-designed playback system (figure 1*a*). Briefly, this system consists of a vibration transducer speaker with a vibrating metal plate (Adin S8BT 26 W, Shenzhen, China) on which we attached (Loctite UltraGel Control, Hemel Hempstead, UK) a plastic base with a metal rod (15 cm tall and 0.5 cm diameter) also glued. The vibration transducer was held with a three-pronged clamp attached to a metal base. We glued (Loctite UltraGel Control) a metal wire (6 cm length and 0.1 cm diameter) to the end of the metal rod, which could then be moved to the desired position. To transmit the vibrations to the flower, a single flower was pinned and glued (Loctite UltraGel Control) on the base of their receptacle perpendicularly to the plane formed by the corolla (figure 1*b*). Finally, the pinned flower was attached to the metal wire of the vibration transducer system using a reusable adhesive (BluTack, figure 1*a*). The system (vibration transducer and attached flower) could then be positioned as required. The definition of x, y and z axes in our study follows Vallejo-Marín [13] and is shown in figure 1*b*.

## 2.4. Measurement of vibrations

Once the flower was positioned, we then deployed two laser Doppler vibrometers (PDV-100, Polytec, Waldbronn, Germany) facing each other (parallel laser beams) and parallel to the axis of the main displacement of the vibration transducer system (figure 1*a*, electronic supplementary material, figure S1). One of the lasers (reference) was always aimed at the flower receptacle, behind the flower, while the other one was aimed either at (i) the pore of the pollinating anther (figure 1*b*), (ii) the pore of one randomly chosen feeding anther (figure 1*c*), or (iii) the third upper quarter along the main vein of the upper petal (figure 1*d*). Since the receptacle of *S. rostratum* is covered by trichomes, we used a small amount (2 to 3 mm$^2$) of reflective tape to improve the quality of the signal reflected to the vibrometer. Both laser vibrometers were connected to a second computer where the mechanical vibration of the receptacle and the other flower structures were simultaneously recorded using the VibSoft-20 software (Polytec, Waldbronn, Germany).

We set the laser vibrometers to a sensitivity of 500 mm s$^{-1}$ and used a low-pass filter of 5 kHz and no high-pass filter. Since vibration signals, especially amplitude, can damp or amplify during their transmission in plant tissues [5], we adjusted the input stimulus to the desired amplitudes (see below) in the flower receptacle using the volume control of the computer and visually checking the amplitude of the reference signal. The amplitude of the input vibration (reference signal in the receptacle) was set as close as possible to one of three amplitude values: 14, 28 or 57 mm s$^{-1}$ root mean-squared amplitude velocity ($V_{RMS}$). The $V_{RMS}$ of the vibrations measured at the receptacle are shown in the electronic supplementary material, table S1. The three target amplitude values correspond to 20, 40 and 80 mm s$^{-1}$ peak amplitude velocity ($V_{PEAK}$), respectively, and are within the range of floral vibrations measured on flowers during buzz pollination by bumblebees. At a frequency of 300 Hz, these peak velocities correspond to peak accelerations of 37.7, 75.4 and 150.8 m s$^{-2}$, and peak displacements of 10.61, 21.22 and 42.44 µm, respectively [13].

Vibrations were simultaneously recorded at both receptacle and other flower structures at a sampling frequency of 12 000 samples per second, 781.25 mHz frequency resolution, using a 0–5 kHz bandpass filter, and recorded for 1.28 s. The input vibrations (at the receptacle) could be calibrated very close (±3 mm s$^{-1}$) to the target amplitude velocities of $V_{RMS} = 14$, 28 and 57 mm s$^{-1}$ (electronic supplementary material, table S1). We played the vibrations at each of the three spatial axes and at each of three amplitude levels in a set of 10 fresh, unvisited *S. rostratum* flowers ($n = 2$ recordings $\times$ 3

floral structures × 3 axes × 3 amplitudes levels × 10 flowers = 540 vibration recordings). One of the measurements (feeding anther, y-axis, input velocity 57 mm s$^{-1}$, flower accession 10-s-77-19) was missed during the recording phase and thus not included in the analysis.

## 2.5. Data analysis

The last 300 ms of each recording was used for analysis. The full recording is shown in electronic supplementary material, figure S2. The power spectrum density (PSD) was estimated using a Hamming window. We estimated the dominant frequency from the PSD. We also estimated the V$_{RMS}$ as a measure of vibration amplitude. For this analysis, we used the packages *seewave* [37] and *tuneR* [38].

To determine how vibration amplitudes are transmitted to different floral structures, we used linear mixed effects models with Gaussian distribution implemented in the R package *lme4* [39]. In these models, the vibration amplitude velocity (V$_{RMS}$) measured in the corolla, feeding or pollinating anthers were the response variable. The input amplitude velocity (V$_{RMS}$) measured at the receptacle, the type of floral structure (corolla, feeding anther or pollinating anther) and the axis of measurement (x, y or z) were considered fixed effects, and plant accession (maternal family) was considered a random effect. We compared models with and without the interaction among fixed effects and a null model considering only the random effect using the Akaike information criterion (AIC) [40]. We compared the estimated ΔAIC (the difference between the AIC for the ith model and the minimum AIC among all the models) of each model to choose the best fit. Values of ΔAIC within 0–2 are considered to have substantial support, within 4–7 considerably less support, and greater than 10 essentially no support [40]. The AIC as well as other model parameters for comparison were estimated using *AICcmodavg* [41]. The best model was used for parameter estimation of the fixed effects and their interaction assessed using *lmerTest* [42]. Model predictions of the fixed effects in these analyses were plotted using the *pred* option in *sjPlot* [43]. All analyses were done in R v. 3.6.2 [44].

## 3. Results

Floral organs (receptacle, petals, feeding and pollinating anthers) mostly presented the same dominant frequency of the input signal (300 Hz), no matter the vibration axes considered (figure 2). Only 2.5% of the estimated dominant frequencies differed from the input frequency (electronic supplementary material, figure S2). The selected statistical model for amplitude velocity included two-way interactions between input velocity and both floral structure and axis of measurement (electronic supplementary material, table S2 and S3). These statistical interactions suggest that differences in amplitude velocity depend on both the floral structure and the axis of measurement. The amplitude velocity (V$_{RMS}$) of vibrations transmitted from the receptacle to the petals closely resembled the input vibration amplitude, showing little evidence of either damping or resonance (table 1 and figure 3). By contrast, the vibrations transmitted to the tip of both feeding and pollinating anthers were consistently of higher amplitude velocity than the input velocity, revealing amplitude increases that ranged from 69% to 443% (table 1, figure 3). The axis with the smallest velocity amplification was the x-axis (69%—110%), while the y- and z-axes showed a higher signal amplification (176%—397% in the y-axis and 229%—443% in the z-axis; table 1, figure 3). Within anthers, both pollinating and feeding anthers showed a very similar relationship with input velocity across the three axes of measurement (figure 3).

## 4. Discussion

Our study contributes to understanding how vibrations are transmitted through floral tissues across multiple spatial axes and on multiple floral organs. We found that the spectral properties of floral vibrations are little affected during their transmission through flowers, with most vibrations retaining their fundamental frequency, and some vibrations, particularly on anthers, being enriched with additional harmonics. Our experiments indicate that the amplitude of floral vibrations (V$_{RMS}$) are magnified up to greater than 400% in anthers and to a much lesser extent in petals. However, this amplification depends on the axis on which the vibrations are applied. Together, our results suggest that floral structures differ in their capacity to transmit vibrations and raise the possibility that the way in which vibrations are applied to the flower (for example, where the vibrations are applied and along which axis) may influence their magnitude and thus their effect on pollen release.

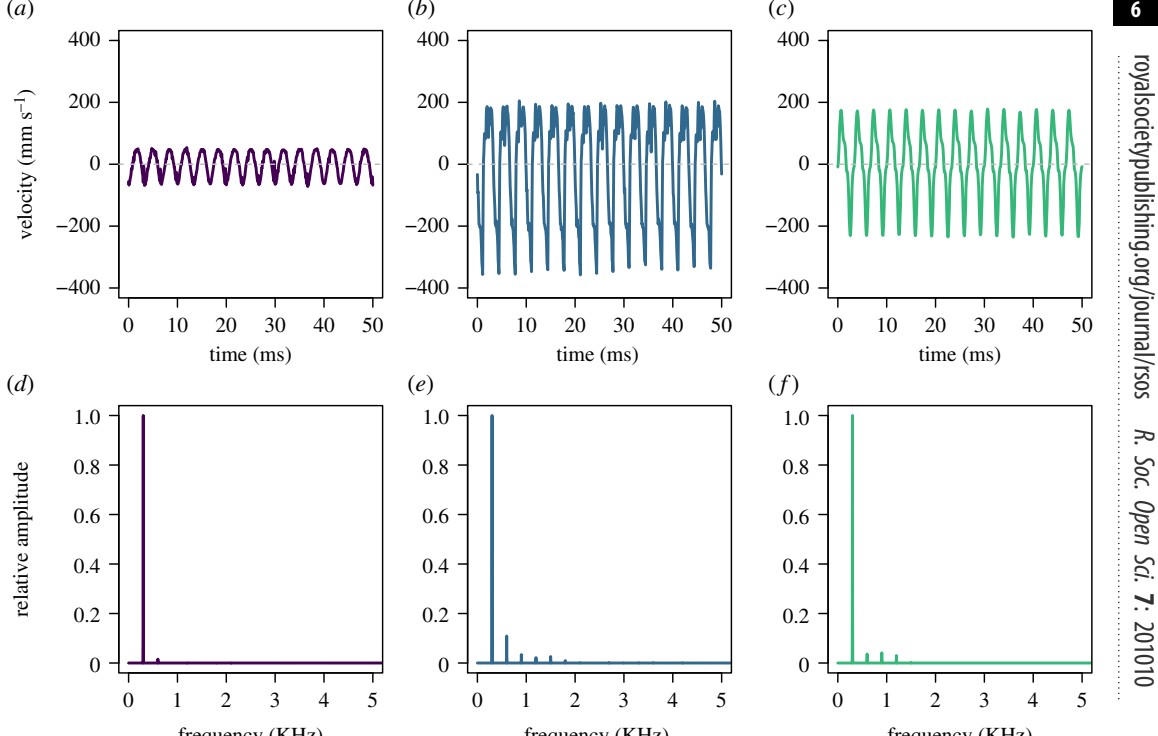

**Figure 2.** Floral vibrations transmitted from the receptacle at the base of the flowers to three different floral organs of the buzz-pollinated plant *S. rostratum*: petal (*a,d*), feeding anther (*b,e*) and pollinating anther (*c,f*). The (*a,b,c*) shows a 50 ms section of the floral vibration in the time domain, and the (*d,e,f*) show the same floral vibration in the frequency domain (power spectrum). All vibrations shown here were measured in flower accession 10-s-77-12, in the y-axis and with an input velocity of $V_{RMS} = 28$ mm s$^{-1}$.

**Table 1.** Transmission of vibrations in buzz-pollinated flowers of *S. rostratum* (Solanaceae). Input vibrations were applied with a mechanical shaker at the base of the flower (receptacle), and measured at one of three floral structures: petals, feeding anthers or pollinating anthers. The vibrations were applied and measured along the same axis (x, y or z; figure 1). Input vibrations had a frequency of 300 Hz, and a RMS amplitude velocity ($V_{RMS}$) of either 14, 18 or 57 mm s$^{-1}$. The table shows the mean $V_{RMS}$ and the 95% confidence intervals (CI; in parenthesis) of measured floral vibrations in each of the three floral structures and axes. The percentage shows the relative change of the measured vibration compared to the targeted input vibration. Positive values indicate amplitude increases. Sample size: 540 vibration measurements from 10 flowers.

| spatial axis | target input velocity $V_{RMS}$ (mm s$^{-1}$) | observed velocity $V_{RMS}$ (mm s$^{-1}$) corolla | feeding anther | pollinating anther |
|---|---|---|---|---|
| x | 14 | 15 (5.9–24.1) 8% | 29.2 (22.2–36.2) 109% | 25.2 (14.1–36.3) 80% |
| | 28 | 30.2 (18.1–42.4) 8% | 53.2 (39.9–66.4) 90% | 45.5 (29.0–62.0) 63% |
| | 57 | 65.7 (37.8–93.7) 15% | 90.8 (78.1–103.6) 59% | 109.8 (68.0–151.6) 93% |
| y | 14 | 19.7 (13.1–26.2) 40% | 69.4 (42.0–96.8) 395% | 60.6 (43.4–77.9) 333% |
| | 28 | 30.8 (23.1–38.6) 10% | 138.8 (86.2–191.5) 396% | 101.5 (71.4–131.7) 263% |
| | 57 | 60.1 (47.1–73.2) 6% | 193.2 (151.7–234.7) 239% | 157.5 (111.8–203.2) 176% |
| z | 14 | 20.1 (13.9–26.2) 43% | 76.3 (53.9–98.8) 445% | 65.6 (34.5–96.7) 369% |
| | 28 | 41.0 (27.9–54.0) 46% | 143.9 (94.8 –193.1) 414% | 115.9 (63.0–168.8) 314% |
| | 57 | 82.1 (60.0–104.2) 44% | 220.1 (179.1–262.7) 288% | 187.4 (122.5–252.4) 229% |

## 4.1. Frequency and amplitude of floral vibrations

We found that the dominant frequency of floral vibrations remains mostly unchanged as it travels through the flower. Similarly, studies of vibration transmission through plant tissues during insect

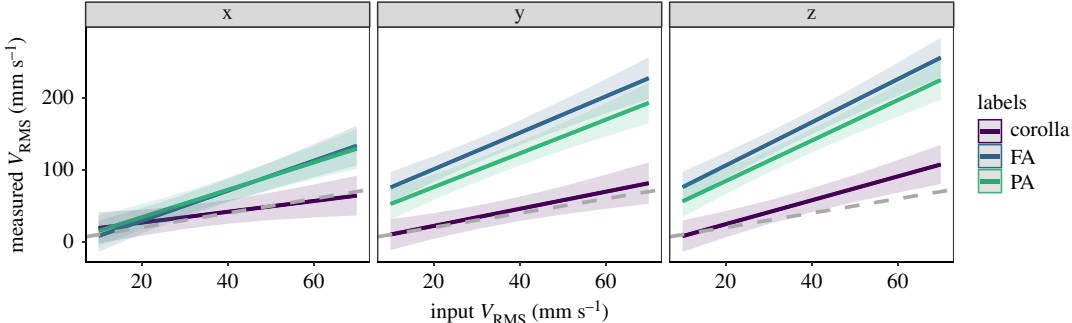

**Figure 3.** Model estimates of input amplitude velocity ($V_{RMS}$) measured at the receptacle, type of floral structure (corolla, feeding anther (FA) or pollinating anther (PA)), and axis of measurement (x, y or z) on the root mean-squared velocity ($V_{RMS}$) of floral vibrations in *S. rostratum*. Lines show the model fit and shadowed areas indicate standard errors. Grey dashed line: receptacle; Purple line: corolla; blue line: feeding anther; green line: pollinating anther. Sample size: 540 vibration measurements from 10 flowers.

communication have also shown preservation of spectral characteristics, which is essential to intra- or interspecific localization and recognition [45–47].

Our finding that floral structures, particularly anthers, amplify the amplitude velocity of input vibrations, while the frequency remains the same, suggests that anther pores are experiencing amplified accelerations [13] and thus higher forces at the anther tips (force = mass x acceleration; [48]). In a cantilever beam, it is expected that the amplitude of oscillation at their tip is positively related to their length [27]. However, we found no difference in the recorded amplitude on different anther types. As these anthers also present different morphologies [28], it is possible that other stamen properties such as stiffness and the second moment of area compensates for differences in anther length, reducing the expected differences in measured amplitude. A very rough estimate of the second moment of area can be obtained using the width of feeding and pollinating anthers of *S. rostratum* measured in a previous study [17], ignoring differences in anther geometry and using a circular cross-section approximation [49]. Pollinating anthers have a greater second moment of area ($I$) than feeding anthers ($I_{PA} = 5.342 \times 10^{-12}$ m$^4$ versus $I_{FA} = 3.137 \times 10^{-13}$ m$^4$). As bending is inversely related to the second moment of area, the higher values of $I_{PA}$ compared to $I_{FA}$ could reduce the displacement of pollinating anthers during vibration, compensating for their longer length compared to the shorter feeding anthers. These data suggest that the similar values of displacement observed in feeding and pollinating anthers could be partly explained by differences in second moment of area that offset length differences between the two anther types.

Despite the amplification of the $V_{RMS}$ measured in the corolla, vibrations transmitted to petals are lower than the vibrations transmitted to the anthers in the same axes. In *S. rostratum*, petals do not vibrate at higher amplitudes along the axis parallel to the central axis of the flower as it would be expected if the petal conformed to a simple cantilever [27]. When predicting displacement of the free end of cantilevers, it is important to consider possible counter forces restricting the bending experienced by the beam as a whole [27]. In *S. rostratum*, the fusion among petals together with the relatively large air-contact and attachment areas may explain the lack of amplitude amplification as well as of amplitude differences among the three spatial axes in the corolla. Differences in biomechanical properties among petals and anthers suggest that vibrations measured in the corolla may not necessarily reflect the vibrations experienced at anther tips.

## 4.2. Bee vibrations

Our study applied vibrations at the base of the flower, which allowed us to study the biomechanical properties of freely moving floral structures. However, during buzz pollination, floral structures are in direct contact with the body of the vibrating bee. For instance, bees tend to grasp the anthers with their mandibles while curling their body around the anthers [12]. Such direct contact between bees and floral structures is likely to influence the vibration properties of the coupled bee–flower system. Further work is required to understand how the interaction between bee and flower during buzz pollination (e.g. where and how the bee holds the flower) affect the vibrations experienced by the anthers. Nevertheless, our study indicates that the axis in which floral vibrations are applied will influence the magnitude of the amplitude velocity experienced at the anther tips. As vibrations of higher amplitude are associated with increased pollen release [33], bees might benefit from exploiting

the biomechanical properties of flowers described here and applying vibrations in the axes associated with higher vibration amplitudes. Previous work has shown that buzz-pollinating bees improve their pollen collection as they gain experience in manipulating flowers [18]. It is possible that such pollen collection improvement involves adjustments on the handling of the flower, including the axis in which the bee applies floral vibrations. If this is the case, we would predict that vibrating bees adjust their position during flower visitation to match the axis of higher vibration amplitude of the anthers (i.e. y and z axes). Future work on how bees manipulate flowers during buzz pollination, particularly how they apply vibrations to anthers, will help elucidate if bees are able to exploit the biomechanical properties of flowers to maximize pollen collection.

Data accessibility. The code as well as the raw data is available from the Dryad Digital Repository at: https://doi.org/10.5061/dryad.f7m0cfxsm [50].

Authors' contributions. V.L.G.B. carried out the laboratory work, participated in data analysis, participated in the design of the study and drafted the manuscript; C.E.P.N. carried out the laboratory work, participated in data analysis, participated in the design of the study and helped draft the manuscript; C.R.R. carried out the laboratory work and participated in the design of the study; F.M.-Z. critically revised the manuscript; M.V.-M. participated in data analysis, participated in the design of the study and helped draft the manuscript. All authors gave final approval for publication and agree to be held accountable for the work performed therein.

Competing interests. We declare we have no competing interests

Funding. We thank FAPEMIG (APQ02497-16) and UFU-CAPES-PrInt (88887.374220/2019-00) grants to V.L.G.B. supporting two visits to the University of Stirling, and a Research Grant from The Leverhulme Trust (RPG-2018-235) to M.V.-M. and F.M.-Z. This research was supported by Scottish Plant Health Licence PH/38/2018-2020.

Acknowledgements. We thank Alistair Gordon for building the vibration playback system (BSc Dissertation, University of Stirling, 2016) and the Vallejo-Marín Laboratory for help with plant growth and discussions on buzz pollination. We thank four anonymous reviewers for their constructive comments on previous versions of the manuscript.

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
