## [Reviewer comments · Royal Society Open Science]

Review History

RSOS-201010.R0 (Original submission)

Review form: Reviewer 1

Is the manuscript scientifically sound in its present form?

Yes

Are the interpretations and conclusions justified by the results?

Yes

Is the language acceptable?

Yes

Do you have any ethical concerns with this paper?

No

Have you any concerns about statistical analyses in this paper?

No

Recommendation?

Accept with minor revision (please list in comments)

Comments to the Author(s)

This is a much improved version. I could not find specific mentioning of sample sizes, please could you add them in the main text as well as in the supplementary tables.

Review form: Reviewer 2**Is the manuscript scientifically sound in its present form?**

Yes

Are the interpretations and conclusions justified by the results?

Yes

Is the language acceptable?

Yes

Do you have any ethical concerns with this paper?

No

Have you any concerns about statistical analyses in this paper?

No

Recommendation?

Accept with minor revision (please list in comments)

Comments to the Author(s)

This paper by Brito et al. investigates how vibrations of floral structures are transmitted between different parts of the flower. This type of vibration occurs when bees visit the flower and mechanically sonicate the flower to release pollen – buzz pollination. By artificially vibrating the base of the flower and measuring the corresponding vibrations in the petals and anthers, the authors find that the frequency of vibrations does not substantially change across different parts of the flower. Additionally, the amplitude of vibrations in the petals is similar to the input vibrations but is greatly increased in the anther tips. The authors suggest that this enhanced amplitude would increase pollen release in the anthers.

The manuscript is well-written, clear and concise with helpful figures that sufficiently illustrate the experimental set up and findings. The methodology is generally sound and the conclusions drawn are appropriate. My only criticism would be that the application of vibrations to the base of the flower is not very ecologically realistic given that bees would approach the anthers. Nevertheless, I appreciate the experimental constraints and difficulty of applying vibration to the anthers themselves.

One further suggestion relates to the finding that the different types of anthers do not appear to show differences in amplitude. The authors suggest that this might be due to differences in stiffness or geometry of the anther types. To further investigate this, the authors could use the geometrical data from their previous paper (ref 28) on these flowers to approximately estimate the second moment of area of the different anther types. This would help to understand whether potential differences in flexural rigidity are likely to be due to geometry or material stiffness. Beyond this, the study is well-designed and carried out and addresses a specific and interesting issue regarding plant-pollinator interactions. The paper will be appreciated by those interested in plant biomechanics, bee pollination and floral morphology.

A few minor points/typos below:

- Paragraph starting in line 62: consider stating at this point that species being studied here is heteroantherous as it is implied but not clarified until the methods section.

- Line 79: 'Does' should be 'Do'
- Line 124: 'on' should be 'of'
- Line 140-141: 'to' should be 'at'
- Line 150: 'specially' should be 'especially'

Some minor changes to the code (removing the author's own directory information from the code) would allow it to be more easily run by others - see comments on the section on data availability.

Review form: Reviewer 3

Is the manuscript scientifically sound in its present form?

Yes

Are the interpretations and conclusions justified by the results?

Yes

Is the language acceptable?

Yes

Do you have any ethical concerns with this paper?

No

Have you any concerns about statistical analyses in this paper?

No

Recommendation?

Accept with minor revision (please list in comments)

Comments to the Author(s)

The present manuscript is a fantastic piece of work studying the vibrational properties of the buzz-pollinated flowers of *Solanum rostratum*. The authors apply different sets of artificial vibrations to single flowers and measure the response vibrations in different floral organs, i.e. petals, feeding and pollinating stamens using laser vibrometry. They find lower vibration resonance in petals than in anthers and conclude that anthers resonate differently than petals, likely an adaptation to optimize pollen release.

This work is the first to demonstrate biomechanical differences in different floral tissues. The manuscript is well written, the combination of methods is highly innovative (i.e. the use of laser vibrometry to measure resonance in flowers) and analyses are sound. I only have a few minor comments.

Comment on the title - the title is very plain now, it could be nice to add that you are studying the transmission of vibrations in heterantherous flowers - I think this makes the story even cooler, because you are dealing with the direct and indirect transmission of vibrations?

49 - insert "to" after "begin"

54 - I would say "focus on"

80 - remove "s" in "difference"

99 - are there bees visiting *S. rostratum* and not buzzing the flowers? Pollen thieves?

124 - should it not say "Shenzhen"?

125 - all that comes in brackets should go after "plate"?

Did you really vibrate for 5 minutes? Does resonance change over time? Is floral tissue wilting in that time?

150 - remove the "s" in "vibrations", and add "e" for "especially"

235 - remove "s" in "others"

246 - in "the" corolla

266 - change "visiting" to "visitation"

Decision letter (RSOS-201010.R0)

Dear Dr Brito

On behalf of the Editors, we are pleased to inform you that your Manuscript RSOS-201010 "Biomechanical properties of a buzz-pollinated flower" has been accepted for publication in Royal Society Open Science subject to minor revision in accordance with the referees' reports. Please find the referees' comments along with any feedback from the Editors below my signature.

Please submit your revised manuscript and required files (see below) no later than 7 days from today's (ie 20-Aug-2020) date. Note: the ScholarOne system will 'lock' if submission of the revision is attempted 7 or more days after the deadline. If you do not think you will be able to meet this deadline please contact the editorial office immediately.

Kind regards,

Anita Kristiansen
Editorial Coordinator

on behalf of Kevin Padian (Subject Editor)
openscience@royalsociety.org

Associate Editor Comments to Author:

Comments to the Author:

Thank you for the engaging manuscript. As you'll see, the referees have a number of suggestions to 'get the paper over the line'. We look forward to receiving your revised manuscript in due course.

Reviewer comments to Author:

Reviewer: 1

Comments to the Author(s)

This is a much improved version. I could not find specific mentioning of sample sizes, please could you add them in the main text as well as in the supplementary tables.

Reviewer: 2

Comments to the Author(s)

This paper by Brito et al. investigates how vibrations of floral structures are transmitted between different parts of the flower. This type of vibration occurs when bees visit the flower and mechanically sonicate the flower to release pollen – buzz pollination. By artificially vibrating the base of the flower and measuring the corresponding vibrations in the petals and anthers, the authors find that the frequency of vibrations does not substantially change across different parts of the flower. Additionally, the amplitude of vibrations in the petals is similar to the input vibrations but is greatly increased in the anther tips. The authors suggest that this enhanced amplitude would increase pollen release in the anthers.

The manuscript is well-written, clear and concise with helpful figures that sufficiently illustrate the experimental set up and findings. The methodology is generally sound and the conclusions drawn are appropriate. My only criticism would be that the application of vibrations to the base of the flower is not very ecologically realistic given that bees would approach the anthers. Nevertheless, I appreciate the experimental constraints and difficulty of applying vibration to the anthers themselves.

One further suggestion relates to the finding that the different types of anthers do not appear to show differences in amplitude. The authors suggest that this might be due to differences in stiffness or geometry of the anther types. To further investigate this, the authors could use the geometrical data from their previous paper (ref 28) on these flowers to approximately estimate the second moment of area of the different anther types. This would help to understand whether potential differences in flexural rigidity are likely to be due to geometry or material stiffness. Beyond this, the study is well-designed and carried out and addresses a specific and interesting issue regarding plant-pollinator interactions. The paper will be appreciated by those interested in plant biomechanics, bee pollination and floral morphology.

A few minor points/typos below:

- Paragraph starting in line 62: consider stating at this point that species being studied here is heteroantherous as it is implied but not clarified until the methods section.
- Line 79: 'Does' should be 'Do'
- Line 124: 'on' should be 'of'
- Line 140-141: 'to' should be 'at'
- Line 150: 'specially' should be 'especially'

Some minor changes to the code (removing the author's own directory information from the code) would allow it to be more easily run by others - see comments on the section on data availability.

Reviewer: 3

Comments to the Author(s)

The present manuscript is a fantastic piece of work studying the vibrational properties of the buzz-pollinated flowers of *Solanum rostratum*. The authors apply different sets of artificial vibrations to single flowers and measure the response vibrations in different floral organs, i.e. petals, feeding and pollinating stamens using laser vibrometry. They find lower vibration resonance in petals than in anthers and conclude that anthers resonate differently than petals, likely an adaptation to optimize pollen release.

This work is the first to demonstrate biomechanical differences in different floral tissues. The manuscript is well written, the combination of methods is highly innovative (i.e. the use of laser vibrometry to measure resonance in flowers) and analyses are sound. I only have a few minor comments.

Comment on the title – the title is very plain now, it could be nice to add that you are studying the transmission of vibrations in heterantherous flowers – I think this makes the story even cooler, because you are dealing with the direct and indirect transmission of vibrations?

49 – insert “to” after “begin”

54 – I would say “focus on”

80 – remove “s” in “difference”

99 – are there bees visiting *S. rostratum* and not buzzing the flowers? Pollen thieves?

124 – should it not say “Shenzhen”?

125 – all that comes in brackets should go after “plate”?

Did you really vibrate for 5 minutes? Does resonance change over time? Is floral tissue wilting in that time?

150 – remove the “s” in “vibrations”, and add “e” for “especially”

235 – remove “s” in “others”

246 – in “the” corolla

266 – change “visiting” to “visitation”

===PREPARING YOUR MANUSCRIPT===

- one version identifying all the changes that have been made (for instance, in coloured highlight, in bold text, or tracked changes);
- a 'clean' version of the new manuscript that incorporates the changes made, but does not highlight them. This version will be used for typesetting.

===PREPARING YOUR REVISION IN SCHOLARONE===

Author's Response to Decision Letter for (RSOS-201010.R0)

See Appendix A.

Decision letter (RSOS-201010.R1)

Dear Dr Brito,

It is a pleasure to accept your manuscript entitled "Biomechanical properties of a buzz-pollinated flower" in its current form for publication in Royal Society Open Science.

on behalf of Professor Kevin Padian (Subject Editor)
openscience@royalsociety.org

Appendix A

Professor Vinícius L. G. Brito
Instituto de Biologia
Universidade Federal de Uberlândia
Brazil

August 26th, 2020

Professor Jeremy Sanders
Journal of the Royal Society Open Science
Editor
University of Cambridge

Dear Prof. Sanders,

We are pleased to submit the revised version of our manuscript entitled “Biomechanical properties of a buzz-pollinated flower” (ID RSOS-201010 by Brito, Nunes, Resende, Montealegre-Zapata, and Vallejo-Marin) that has been accepted pending minor revisions as an original Research Article in the Journal of the Royal Society Open Science (RSOS). All the minor changes were accepted and a detailed response letter in which all reviewers’ comments were addressed (in blue) is given bellow.

I confirm that this contribution is original and that its preprint was previously published in the BioRxiv server (18 March 2020; www.biorxiv.org/content/10.1101/2020.03.17.995746v1).

Please do not hesitate in contacting me if I can be of further assistance.

Sincerely,

Professor Vinícius L. G. Brito
Instituto de Biologia

Response to the reviewers

Associate Editor Comments to Author:

Comments to the Author:

Thank you for the engaging manuscript. As you'll see, the referees have a number of suggestions to 'get the paper over the line'. We look forward to receiving your revised manuscript in due course.

Reviewer comments to Author:

Reviewer: 1

Comments to the Author(s)

This is a much improved version. I could not find specific mentioning of sample sizes, please could you add them in the main text as well as in the supplementary tables.

Reply – We mention sample size in the main text, lines 167 – 172. We have now also added sample sizes in legends of figures, tables and supporting material.

Reviewer: 2

Comments to the Author(s)

This paper by Brito et al. investigates how vibrations of floral structures are transmitted between different parts of the flower. This type of vibration occurs when bees visit the flower and mechanically sonicate the flower to release pollen – buzz pollination. By artificially vibrating the base of the flower and measuring the corresponding vibrations in the petals and anthers, the authors find that the frequency of vibrations does not substantially change across different parts of the flower. Additionally, the amplitude of vibrations in the petals is similar to the input vibrations but is greatly increased in the anther tips. The authors suggest that this enhanced amplitude would increase pollen release in the anthers.

The manuscript is well-written, clear and concise with helpful figures that sufficiently illustrate the experimental set up and findings. The methodology is generally sound and the conclusions drawn are appropriate. My only criticism would be that the application

of vibrations to the base of the flower is not very ecologically realistic given that bees would approach the anthers. Nevertheless, I appreciate the experimental constraints and difficulty of applying vibration to the anthers themselves.

R – We thank the reviewer for the positive and encouraging comments about our manuscript. Our main aim in this study was to describe the vibrational properties of the flower and not of the coupled bee-flower system. Applying controlled mechanical vibrations as a bee would do onto a flower in a more realistic way as suggested by the reviewer would be extremely challenging but is something that hopefully we will try in the future.

One further suggestion relates to the finding that the different types of anthers do not appear to show differences in amplitude. The authors suggest that this might be due to differences in stiffness or geometry of the anther types. To further investigate this, the authors could use the geometrical data from their previous paper (ref 28) on these flowers to approximately estimate the second moment of area of the different anther types. This would help to understand whether potential differences in flexural rigidity are likely to be due to geometry or material stiffness.

R – We thank the reviewer for this great suggestion. We have followed this recommendation and used anther measurements obtained from previous work on *Solanum rostratum* (reference 17) to estimate the second moment of area in pollinating and feeding anthers. We report this calculation in the Discussion, and conjecture on how the higher second moment area of pollinating anthers compared to feeding anthers could compensate for their length differences causing both anther types to have similar vibration amplitudes (lines 238-247).

Beyond this, the study is well-designed and carried out and addresses a specific and interesting issue regarding plant-pollinator interactions. The paper will be appreciated by those interested in plant biomechanics, bee pollination and floral morphology.

A few minor points/typos below:

- Paragraph starting in line 62: consider stating at this point that species being studied here is heteroantherous as it is implied but not clarified until the methods section.

R – As suggested, we have added here that the species studied here is heterantherous (lines 62-63 and line 76).

- Line 79: 'Does' should be 'Do'

R – Changed.

- Line 124: 'on' should be 'of'

R – Changed.

- Line 140-141: 'to' should be 'at'

R – Changed.

- Line 150: 'specially' should be 'especially'

R – Changed.

Some minor changes to the code (removing the author's own directory information from the code) would allow it to be more easily run by others - see comments on the section on data availability.

R – We changed the code accordingly. We have not been able to see additional comments on the “section on data availability” stated by the reviewer.

Reviewer: 3

Comments to the Author(s)

The present manuscript is a fantastic piece of work studying the vibrational properties of the buzz-pollinated flowers of *Solanum rostratum*. The authors apply different sets of artificial vibrations to single flowers and measure the response vibrations in different floral organs, i.e. petals, feeding and pollinating stamens using laser vibrometry. They find lower vibration resonance in petals than in anthers and conclude that anthers resonate differently than petals, likely an adaptation to optimize pollen release. This work is the first to demonstrate biomechanical differences in different floral tissues. The manuscript is well written, the combination of methods is highly innovative (i.e. the use of laser vibrometry to measure resonance in flowers) and analyses are sound. I only have a few minor comments.

R - Thank you for the very positive evaluation of our manuscript.

Comment on the title – the title is very plain now, it could be nice to add that you are studying the transmission of vibrations in heterantherous flowers – I think this makes the story even cooler, because you are dealing with the direct and indirect transmission of vibrations?

R – We have added “heteranthery” to the key words, but we would like to respectfully request to keep the title as in the previous version. We fully agree that an exciting aspect of the study system is the presence of stamen dimorphism within flowers, but believe that adding the botanical term “heteranthery/heterantherous” might provide a language barrier for the majority of readers not already familiar with this peculiar term of the botanical reproductive biology literature. We describe the presence of anther dimorphism in the studied flowers throughout the text so that the readers can familiarize themselves with the meaning of the term before it is introduced. If the editor still considers this necessary, we will be glad to modify the title.

49 – insert “to” after “begin”

R – Changed.

54 – I would say “focus on”

R – Changed.

80 – remove “s” in “difference”

R – Removed.

99 – are there bees visiting *S. rostratum* and not buzzing the flowers? Pollen thieves?

R – The reviewer is correct. Non-buzzing bees visit flowers of *S. rostratum* (eg *Apis mellifera*). Pollen theft is also common in natural populations of this species. Both of these observations are reported in a previous study by one of the authors (Solís-Montero L., C. Vergara, and M. Vallejo-Marín. 2015. High incidence of pollen theft in natural populations of a buzz-pollinated plant. *Arthropod-Plant Interactions*. 9: 599-611. doi:10.1007/s11829-015-9397-5).

124 – should it not say “Shenzhen”?

R – Changed.

125 – all that comes in brackets should go after “plate”?

R – The reviewer is correct. Changed as suggested.

Did you really vibrate for 5 minutes? Does resonance change over time? Is floral tissue wilting in that time?

R – Apologies for the confusion, we only played back 1.28s of the vibration for each floral recording. We have modified the text to try to clarify this by deleting the 5 minute reference and instead leaving only the amount of time actually played back to the flower (1.28 s; line 165).

150 – remove the “s” in “vibrations”, and add “e” for “especially”

R – Changed.

235 – remove “s” in “others”

R – Changed.

246 – in “the” corolla

R – Changed.

266 – change “visiting” to “visitation”

R – Changed.